# The Efficiency of Polyester-Polysulfone Membranes, Coated with Crosslinked PVA Layers, in the Water Desalination by Pervaporation

**DOI:** 10.3390/membranes14100213

**Published:** 2024-10-07

**Authors:** Izabela Gortat, Jerzy J. Chruściel, Joanna Marszałek, Renata Żyłła, Paweł Wawrzyniak

**Affiliations:** 1Faculty of Process and Environmental Engineering, Lodz University of Technology, Wólczańska 213, 93-005 Łódź, Poland; izabela.gortat@dokt.p.lodz.pl (I.G.); pawel.wawrzyniak@p.lodz.pl (P.W.); 2Łukasiewicz Research Network-Lodz Institute of Technology, Circular Economy Center (BCG), Brzezińska 5/15, 92-103 Łódź, Poland; jerzy.chrusciel@lit.lukasiewicz.gov.pl (J.J.C.); renata.zylla@lit.lukasiewicz.gov.pl (R.Ż.)

**Keywords:** composite membranes, active layer, crosslinked poly(vinyl alcohol), glutaraldehyde, citric acid, pervaporation, saline water, desalination

## Abstract

Composite polymer membranes were obtained using the so-called dry phase inversion and were used for desalination of diluted saline water solutions by pervaporation (PV) method. The tests used a two-layer backing, porous, ultrafiltration commercial membrane (PS20), which consisted of a supporting polyester layer and an active polysulfone layer. The active layer of PV membranes was obtained in an aqueous environment, in the presence of a surfactant, by cross-linking a 5 wt.% aqueous solution of polyvinyl alcohol (PVA)—using various amounts of cross-linking substances: 50 wt.% aqueous solutions of glutaraldehyde (GA) or citric acid (CA) or a 40 wt.% aqueous solution of glyoxal. An ethylene glycol oligomer (PEG 200) was also used to prepare active layers on PV membranes. Witch its help a chemically cross-linked hydrogel with PVA and cross-linking reagents (CA or GA) was formed and used as an active layer. The manufactured PV membranes (PVA/PSf/PES) were used in the desalination of water with a salinity of 35‰, which corresponds to the average salinity of oceans. The pervaporation method was used to examine the efficiency (productivity and selectivity) of the desalination process. The PV was carried at a temperature of 60 °C and a feed flow rate of 60 dm^3^/h while the membrane area was 0.005 m^2^. The following characteristic parameters of the membranes were determined: thickness, hydrophilicity (based on contact angle measurements), density, degree of swelling and cross-linking density and compared with the analogous properties of the initial PS20 backing membrane. The physical microstructure of the cross-section of the membranes was analyzed using scanning electron microscopy (SEM) method.

## 1. Introduction and Short Review

The global water shortage is becoming an increasingly significant problem for humanity. The changing climate caused by human activities, deforestation, and artificial drainage of land leads to a growing disruption of the global water balance [1,2,3,4]. According to reports from the World Health Organization (WHO), the United Nations Educational, Scientific and Cultural Organization (UNESCO), and the World Wide Fund for Nature (WWF), climate change is causing and will cause more and more frequent and more severe periodic water shortages [5,6,7,8,9]. The solution to the problem of obtaining a fresh water, both for industrial and food purposes, is the use of the sea water in the desalination process, which creates an important motivation for the work of scientists [10,11,12]. Initially, desalination methods were based on its thermal separation, i.e., distillation. Thermal methods have been successfully used for the desalination of sea and brackish water [13,14,15,16]. Seawater has been successfully desalinated since the 1960s using membrane techniques. The most popular membrane methods for the water desalination include primarily the reverse osmosis (RO) process and filtration methods, in particular nanofiltration (NF). There has been also observed a growing interest in processes such as forward osmosis, membrane distillation (MD), and the application of hybrid methods, like the nanofiltration and reverse osmosis (NF-RO) [15,17,18,19,20].

The pervaporation (PV) process as a modern separation membrane technique for special tasks, and in particular for the water desalination, is relatively little known. Currently, studies are being conducted on this process and the possibilities of its applications for obtaining desalinated water [21,22]. The pervaporation is a low-pressure membrane separation technique that uses the difference in solubility of individual components of the separated mixture inside the membrane. The mechanism of mass transport in the pervaporation process is based on the solution–diffusion model. The membranes used for this process are asymmetric with a active layer in which the adsorption, dissolution of the solution components and diffusion through the polymer material take place, followed by the permeation of the evaporated components through the lower (support) layer their transport. In the next stage, desorption and condensation into the liquid phase take place [23,24,25,26]. However, a significant limitation is a lack of appropriate membranes on the global market that are reliable, durable, highly efficient and selective. In recent years, intensive research has been carried out on the production of membranes for the PV process, both containing additional components in the form of silica, zeolites, metal oxides or graphene, as well as membranes without the use of micro- and nanofillers [22,27,28,29,30].

The aim of this study was to develop polymeric membranes intended for the desalination of saline water which will achieve high efficiency and selectivity in PV process. Pervaporation mem-branes were fabricated by modifying the surface of polyester-polysulfone membranes and covering them with an active layer of cross-linked polyvinyl alcohol. Process efficiency is determined by the permeate stream. Selectivity describes the degree of salt retention. In the literature, there are examples of the use of polymer PV membranes in the water desalination process. Comparing them is difficult due to the different thicknesses of the membranes used, different brine concentrations expressed in the amount of NaCl and different conditions of the PV process (temperature and pressure on the low pressure side of the membrane) [20].

The present studies focused on verifying the suitability of various crosslinking substances in forming an active layer PV membrane. The produced active layer mod-ified the properties of a commercial PS20 polyester-polysulfone bilayer UF membranes so that they became effective in the pervaporation desalination of water. The influence of cross-linking reagents on the structure of the modified active layer of the obtained membranes, the influence of the type of surfactant on the transport properties of the PV membrane and the effect of the PVA content on the properties of the modified supporting PS20 membrane were determined. Comparison of the properties of the modified fabricated PV membranes allowed us to develop the most favorable recipes for a surface modification of the polymer matrix in active and support layer.

### 1.1. Materials and Methods Production of Polymer Membranes

The most popular methods of obtaining polymer membranes include: electrospinning, phase inversion and dip coating [12]. Phase inversion involves pouring the membrane-forming solution onto a previously prepared composite polymer primer. Subsequently, the composite membrane coated with the polymer solution is most often immersed in water (or in other solvent), which leads to precipitation of the polymer from membrane-forming solution film (this is the so-called “wet phase inversion”). Next a chemical cross-linking can occur with a cross-linking reagent, if it is present in the membrane forming solution. Alternatively, a polymeric surface layer can be formed by heating the membrane at appropriate temperature (by the so-called “dry phase inversion”). The phase inversion method is suitable for producing flat membrane sheets due to the possibility of adjusting the thickness of the modified membranes. Immersion and electrospinning methods are successfully used to produce both flat membranes and the so-called hollow fibers, due to the possibility of obtaining thinner layers applied to tubular membranes [12,23,28,31,32,33,34,35,36,37,38].

Polymer membranes for the pervaporation process are referred to as composites because they are multilayer and multicomponent systems of cross-linked polymers, containing different additives (fillers, etc.) and can be prepared by various recipes. Cheng et al. [26] described the use of a number of polymeric materials, including: polyvinylpyrrolidone, polyvinyl acetate, cellulose acetate as well as recycled materials, e.g., cellulose—for the production of membranes useful in the pervaporation process.

Due to the diffusive nature of separation and the significant pressure difference in the pervaporation process, producing membranes with satisfactory efficiency and selectivity is a great challenge. On the one hand, a membrane must be produced whose active layer is solid (i.e., non-porous on the surface), and on the other hand it must have high mechanical strength and chemical resistance during operation. The method of cross-linking the polymer inside the active layer membranes is important. Densely cross-linked polymers retain ions on the surface or inside the membrane, preventing their transport. An insufficient degree of cross-linking, in turn, may cause the polymer to wash out, which results in the disruption of the membrane structure [24]. A crucial challenge in the preparation of membranes is the use of a proper polymer that is appropriate in terms of miscibility, compatibility and properties, constituting the most important ingredient of the membrane-forming solution or dispersion. Polyvinyl alcohol (PVA) is an excellent polymer due to its hydrophilic and biodegradable properties and is successfully used to produce polymer membranes for water desalination. PVA is highly soluble in water and forms hydrogels, and its cross-linking, e.g., with dicarboxylic acids [24], is not a major problem. Most often, PVA was cross-linked with glutaraldehyde, but this reagent is increasingly being abandoned due to its carcinogenic properties [24,39,40,41,42,43]. Truong et al. [44] applied electrospinning method for the preparation of PVA-based polymer membranes, which were further cross-linked with citric or maleic or polyacrylic acid.

### 1.2. Polymer Membranes Used for Pervaporative Water Desalination

At present the search for new recipes to produce polymer membranes for water purification is an important area of scientific research. Wang et al. [25] paid particular attention to the pervaporation process, which is competitive with the reverse osmosis (RO) process. Wang et al. [27] also described the use of the pervaporation process for water desalination. The detailed classification of membranes for the PV process was provided by Liu et al. [45] dividing them into organic, inorganic, polymer, 2D and mixed-matrix membranes (MMM). A the same time, the analysis of membrane compositions for other membrane processes (RO and NF) appears to be a suitable approach for achieving satisfactory results in the preparation of polymer membranes for the PV process.

Table 1 presents an overview of the work related to the production of pervaporation membranes for water desalination. It shows different production methods, different materials and initial results. The authors in their studies did not demonstrate the full efficiency of PV membranes. Analyzing the various possibilities for membrane fabrication leads to the conclusion that this is an area with significant potential for benefiting industrial applications.

In the preparation processes of PV membranes, attention should be paid to the types of additional fillers used, which can cause changes in the performance or selectivity of the membranes. Zeolites represent broad group of fillers used in the preparation of PV membranes. Yu et al. [52] utilized zeolites with CHA and FAU structures to produce polymer PV membranes for desalination processes. Ge et al. [53] reviewed the available methods for the fabrication of pervaporation and nanofiltration membranes containing graphene oxide (GO) additives.

Prihatiningtyas et al. [54] described various types of nanocomposite PV membranes for water desalination. They focused on organic and inorganic membrane types as well as hybrid systems. The performance and selectivity of selected membranes were also discussed. Alkhouzaam et al. [55] reviewed the use of GO for water desalination and wastewater treatment. Using functionalized and modified GO they achieved higher efficiency and selectivity for TFN and TFC membranes, compared to membranes with mixed matrix (MMM). At the same time, MMM membranes showed high capabilities in retaining organic particles and heavy metals.

Zeng et al. [56] emphasized challenges in developing structurally efficient membranes containing GO due to π–π interactions. They designed a method to graft PVA onto GO and obtain the PVA-GO nanocomposite, which ensured better dispersion of GO and membrane hydrophilicity. PAN substrates were used. It was shown that the use of GO in PVA-GO composite membranes increased separation efficiency and improved anti-swelling properties. Saleem et al. [57] described the use of membranes containing microporous carbon nitrides (g-C_3_N_4_, C_2_N, and C_3_N) in various water desalination processes, including pervaporation.

According to Singha et al. [58], membrane processes are an excellent solution for obtaining purified water and have the potential to address the global water scarcity problem. The authors described a variety of membrane types that can successfully be used in desalination and water purification processes using the unconventional method of pervaporation. They emphasized the significance of the highly commercialized RO method, which has limitations but also potential for desalination in the PV process. The authors identified the PV process as having a higher potential for energy efficiency. By employing hydrophilic membranes, the PV process has an advantage over membrane distillation due to its capability to handle high concentrations of substances in the feed while simultaneously absorbing the solvent into the membranes.

The methods of membrane fabrication can impact their separation properties. Athayde et al. [59] described a two-step process for manufacturing membranes, involving wet and dry coating of a carbonaceous membrane surface. Initially, the substrate pores were wetted and then immersed in phenolic resin, followed by a second coating layer and carbonization at temperatures ranging from 600 to 800 °C. They demonstrated that higher carbonization temperatures had a beneficial effect on achieving higher flux values of purified water. The water desalination results provided substantial evidence that carbon membranes obtained by the two-step method allowed high salt rejection performance [59].

Table 2 contains a literature review extended to include studies on the efficiency of feed desalination in the pervaporation process on manufactured membranes. It is worth noting that the works mentioned were created within the last few years. This confirms the importance of the topic discussed in this study and the innovative nature of the research performed. Currently, research is focused on developing innovative membranes for water desalination processes. Li et al. [21] also conducted a review of available membranes for the PV process, providing a detailed description of the polymer materials and additives used in the preparation of PV membranes dedicated to water desalination. This review serves as a foundation for exploring the most favorable formulations and the application of specific polymer and crosslinking agent combinations, from which the authors of this work drew their insights. The process temperature of 60 °C was used, similar to Li et al., and the most favorable polymer, PVA, was utilized in the studies.

From the literature analysis in Table 2, it is also evident that the use of diverse composite membranes containing PVA as a precursor for the active polymer matrix in pervaporation membranes yields satisfactory or even remarkable results—see the last column of Table 2 *J_p_* = 124.8 kg/(m^2^·h). These desalination performances of PV membranes are the best reported so far for PV composite membranes with a NIPS-prepared substrate, demonstrating their potential for industrial applications [73].

Solutions containing PVA can successfully be applied onto various substrates, including synthetic polymer-based ones like PAN, as well as natural polymers such as lignin. PVA can serve as a component for the active layer of membranes made from different polymers. The use of cross-linking agents with different chemical structures, such as glutaraldehyde, citric acid, glyoxal, tannic acid, or maleic acid, is beneficial for a membrane-forming system containing PVA, enhancing its separation properties and process efficiency. Moreover, the PV method can be employed for concentrate desalination after the RO process, as described by Li et al. [21] and Zhang et al. [71]. This study highlights the immense potential of pervaporation processes for desalination.

### 1.3. Surfactants in the Preparation of PV Polymeric Membranes

An important step in the process of producing desalination membranes is the control of their hydrophilicity. High surface hydrophilicity reduces the process of membrane fouling [24]. Surfactants are a group of ionic or non-ionic surfactants that play a dual role in polymer membranes. Firstly, they cause an increase in the hydrophilicity of the membrane surface and a greater affinity for water, thus increasing the filtration rate and the degree of water desalination by the membranes. Secondly, surfactants exhibit the so-called antifouling effect, i.e., they have the positive effect on the life of membranes by reducing the deposition of contaminants on their surface, i.e., they limit the so-called fouling [74].

Lau et al. [75] studied the effect of surfactants in the preparation process of polymeric membranes. Surfactants increase the surface wettability, which contributes to better adhesion of various compounds on the membrane surface. According to Jegal et al. [76], the use of triethylbenzylammonium bromide (TEBAB) improved the properties of composite membranes, resulting in increased process efficiency. On the other hand, Mansourpanah et al. [77] demonstrated that the anionic surfactant sodium dodecyl sulfate (SDS) caused defects and cracks on the surface of developed membranes, whereas the cationic surfactant cetyltrimethylammonium bromide (CTAB) increased monomer diffusion during the polymerization process. Meanwhile, the non-ionic surfactant Triton X-100 positively influenced the formation of a denser polymer network in the membrane’s surface layer.

The effect of surfactant concentration on membrane performance was described by Saha et al. [78]. They tested membranes with the addition of sodium lauryl sulfate (SLS) at concentrations ranging from 0.1% to 0.5% by weight. Initially, the performance and selectivity of the membranes remained unchanged, but increasing the SLS concentration to 0.5% by weight resulted in a reduction in salt rejection by up to 10%. A salt solution with a concentration of 2000 ppm was tested. The membrane performance did not deteriorate. It was concluded that the use of the surfactant reduced the surface tension of the material, which in turn led to easier diffusion of salt ions through the membrane. Tsai et al. [79] reached similar conclusions. Additionally, they described the effect of the type of surfactant used on the formation of macropores inside the PSf membrane. Hydrophilic surfactants cause the formation of channels in the hydrophilic membrane-forming system, while lipophilic surfactants do so in lipophilic systems. This has significant implications for the internal structure of the polymer network. Thus, it is possible to control the number of formed channels. As the amount of surfactant increased, the separation factor in the PSf membrane also increased, while the permeability coefficient decreased. The authors, while developing the membrane using wet phase inversion, concluded that the use of a hydrophobic surfactant with low affinity to the coagulant reduced the formation of macropores. Membranes containing the surfactant exhibited significantly higher flux compared to membranes without the surfactant.

Surfactants can have gelling properties similar to polyethylene glycol (PEG). The use of sodium dodecyl sulfate (SDS) in the preparation of polyethersulfone (PES) membranes was described by Alsari et al. [80]. Polyethersulfone (PES) membranes were prepared using solution casting at temperatures of 4 °C and 20 °C. The effect of SDS and the range of molecular weight cut-off (MWCO) of the prepared membranes on transport properties, pore size, and surface tension was investigated. The pore size was larger for membranes prepared at 20 °C. In general, it was found that both the molecular weight cut-off (MWCO) of the polymer and the pore size decreased with an increase in SDS concentration in the gelling medium. The roughness of the membranes increased with the molecular weight cut-off (MWCO) of the polymer and the pore size.

The use of surfactants in the membrane preparation process has many beneficial features important for developing highly efficient and selective membranes. Surfactants not only reduce surface tension but also play a significant role in the formation of channels within the membranes. For this reason, the authors of this study decided to investigate the effect of selected nonionic surfactants on the process of water desalination using the pervaporation (PV) method.

## 2. Materials and Methods

### 2.1. Materials

The outer active layer of PV membrane was cast from an aqueous solution of PVA onto a commercial PS20 membrane (Sepro Membranes Inc., Carlsbad, CA, USA). PS20 was a bilayer ultrafiltration membrane made of polysulfone-polyester with a molecular weight cut-off (MWCO) of 20 kDa. PVA, with a weight-average molecular weight of ~27,000 g/mol and a hydrolysis degree of 99.8%, was a product from Sigma Aldrich (Saint Louis, MO, USA). Aqueous solutions of glutaraldehyde (GA) (Chempur, Piekary Śląskie, Poland) and citric acid (CA) at concentrations of 50% wt. and 40% aqueous glyoxal solution (Merck KGaA, Darmstadt, Germany) were used for crosslinking the PVA. Alternatives to citric acid are tartaric acid and tannic acid (Figure 1) Warchem, Zakręt, Poland. First Tween 20 (product from Thermo Fischer Scientific, Rockford, IL, USA) was used as a surfactant with wetting and antifouling properties. Next, a nonionic detergent LC with a dynamic viscosity in the range of 1080–1200 cP (“Ludwik cytrynowy”, INCO Group, Warsaw, Poland), commonly used in household chemistry was applied. Occasionally two polyethers from PCC Rokita SA (Brzeg Doly, Poland): Rokopol 30P10 (a nonionic surfactant, PEG and PPG copolymer, 10:30 mol/mol) and Rokanol L4P5 (anionic surfactant, diethanolamine salt of polyoxyethylene derivative of lauryl alcohol phosphate) were also used. 5N sulfuric acid (VI) (Chempur, Piekary Śląskie, Poland) was used as a catalyst for the PVA crosslinking reaction. In some experiments, liquid poly(ethylene glycol) (PEG) was additionally used to crosslinking with PVA and preparing the hydrogel active layer of PV membrane. PEG 200 was a product of PCC Exol SA (Brzeg Dolny, Poland) with an number average molecular weight (M_n_) of 190–210 g/mol. For pervaporation processes, an aqueous solution of sodium chloride NaCl (Chempur, Piekary Śląskie, Poland) with the concentration of 3.5% wt., corresponding to the average salinity of oceans, was prepared as the feed, according to the literature [81].

Pervaporation composite PVA/PSf/PES membranes were fabricated by modifying the surface of PS20 membrane. Then PS20 was coated with an active layer of cross-linked PVA. To obtain a layer of uniform thickness (50 µm) the poured layer of PVA solution was evenly spread using Adjustable Baker Film Applicator (Elcometer model 3530/5, Warren, MI, USA). The membrane prepared in this way was then dried in a laboratory dryer, initially at 80 °C for 30 min, and then at 105 °C for 15 min.

### 2.2. Research Methods

#### 2.2.1. Analytical Methods

The membranes were subjected to tests in order to accurately determine their characteristics through measurements of thickness, contact angle, and swelling degree. Membrane thickness affects its transport properties, such as membrane resistance, which can impact the efficiency of the PV process and mechanical stability. The membrane PVA/PSf/PES thickness was measured using a micrometer sensor (Hogetex 9M02.2.07, Nieder-Olm, Germany). The determination of the active layer thickness involved calculating the difference between the average thickness of the produced membranes and the thickness of the PS20 substrate membrane. SEM images (Jeol JSM-6360, Tokyo, Japan) of selected membranes were taken to reveal the surface structure characteristics of the active layer of the membrane.

The hydrophilicity of membranes, which indicates the affinity of the active layer surface to water, was determined using a goniometer (DSA 25 Kruss, Hamburg, Germany). Due to the necessity of achieving hydrophilic surfaces in membranes used for pervaporation desalination of water, the contact angle is a significant parameter analyzed during membrane preparation.

The swelling degree (S), which indicates the affinity of the material to water, was calculated using Equation (3):(1)$$S=\frac{m_{w}-m_{d}}{m_{d}}\cdot 100,\left[\%\right]$$
where: *m_d_* is the mass of the dry membrane, and *m_w_* is the mass of the wet membrane.

The weighed dry membrane was conditioned in demineralized (DI) water for a period of 24 h, and then weighed. The membrane saturated with the solvent (water) was referred to as “wet”. The percentage degree of swelling of the active layer of the membrane was calculated as the difference between the degree of swelling of the entire PV membrane and its support layer, which was a commercial PS20 membrane of uniform thickness.

Calculations of membrane crosslinking density were conducted based on the data needed to determine the swelling degree. The crosslinking density (υ) values of the membranes were calculated using Equations (2) and (3):(2)$$\Phi =\frac{\frac{m_{d}}{\rho_{d}}}{\frac{m_{d}}{\rho_{d}}+\frac{m_{w}-m_{d}}{\rho_{H_{2}O}}}$$
(3)$$\upsilon =\frac{{E\Phi }^{-\frac{1}{3}}}{3R\;T}$$
where *ρ_d_* is the density of the dry membrane, *E* is the Young’s modulus of the membrane sample, *Φ* is the volume fraction of the polymer in the membrane.

The crosslinking density was calculated based on the known masses of the membranes and their tensile strength (*E*) [82]. Initially, the volume fraction of the polymer in the membrane (*Φ*) needs to be calculated, determined from the difference between the masses of the wet and dry membranes and their density. Only after determining the tensile strength is the crosslinking density calculated.

An important parameter is the density of the material PV membrane due to the integrity of both layers. The density is was determined for the entire volume of the material, not just the created matrix of the active layer. Experimentally, it was determined based on the amount of displaced fluid during the immersion of the material.

#### 2.2.2. Pervaporation Process Setup

The desalination experiments using the pervaporation (PV) method were conducted using laboratory equipment from Sulzer Chemtech (Allschwil, Switzerland). A flat membrane with an area of A = 0.005 m^2^ was used in the study.

Based on previous research [83,84], the desalination process using the PV method was conducted at a temperature of 60 °C, with a feed flow rate (salted water) (*Q_f_*) of 60 dm^3^/h, and the pressure on the low-pressure side of the membrane was constant at 3 kPa (2.25 Torr). During previous PV desalination studies of model aqueous solutions on commercial membranes: PEVAP 4510, PERVAP 2201, 2202, at the model feed flow rate of 60 dm^3^/h, we did not observe concentration polarization significantly affecting the efficiency of the pervaporation process [85]. For comparison in the experiments shown in the manuscript, we repeated the adopted process conditions. The schematic of the PV process apparatus is shown in Figure 2. In our study, we focused mainly on the influence of individual components on the preparation and properties of the active layer of PV membranes for water desalination. The feed (i.e., a 3.5% NaCl solution) was heated using a thermostat and then fed into the membrane module, which was connected to a vacuum pump. Due to its separation properties, i.e., affinity to the separated component, the membrane selectively transported chosen components of the mixture to the other side of the membrane or retained them in the retentate. The separation process of the salt from water NaCl solution can be explained based on the dissolution-diffusion model. Inside the membrane, there was a phase change of the permeating component from liquid into vapor. The permeating component refers to the component passing through the membrane, which simultaneously changes its state of matter during the separation process. In this case, water vapor (permeate) was condensed in a receiver cooled with liquid nitrogen.

The efficiency of the PV process is determined by the permeate flux (*J_p_*) and is expressed as the amount of obtained permeate (*m_p_*) per unit membrane area (*A*) and process time (*t*) (Equation (4)). The selectivity, described by the degree of salt retention (*R*), was indirectly determined by the conductivity of ions in the permeate (*C_p_*) and filtrate (*C_f_*) (Equation (5)), measured with an Elmetron CPC-511 conductivity meter (Zabrze, Poland).
(4)$$J_{p}=\frac{m_{p}}{A\;t}\;,\;\left[\frac{kg}{m^{2}\;h}\right]$$
(5)$$R=\left(1-\frac{C_{p}}{C_{f}}\right)\cdot 100,\;\left[\%\right]$$

## 3. Results and Discussion

In the experiments on modifying the surface properties of polymer membranes intended for desalination of saline water by pervaporation, the focus was primarily on the impact of the amount of polymer forming the active layer (PVA), the type of crosslinking agents, and the surfactant, as well as their concentrations, on the filtration properties of the membranes. Since the separation studies in this article were conducted on a model system (3.5% wt. NaCl in water), the fouling phenomenon was not observed and no fouling tests were conducted. Fouling or rather scaling was observed in studies of sea and geothermal waters [84].

### 3.1. The Influence of PVA Concentration and Crosslinking Agents on Membrane Preparation

The literature describes the use of various concentrations of PVA for producing membranes for the PV process [24,40,43,85,86,87]. Jayaramudu et al. [88] used a cellulose suspension in a 3 wt.% aqueous PVA solution. In contrast, Zhang et al. [87] applied a 10 wt.% aqueous PVA solution with a degree of polymerization (DP) of approximately 1700 g/mol (corresponding to a number average molecular weight of about 74,800 g/mol), which significantly increased the viscosity of the membrane-forming solution, crosslinked using glutaraldehyde (GA).

In our preliminary studies, the concentrations of aqueous PVA solution were 2%, 4%, and 5% by weight. The viscosity of the solution increases with higher PVA concentrations. The concentration of PVA used to form the active layer of the composite PV membrane affects spreadability of the active layer forming solution, and the wetting of the underlying polyester-polysulfone (PS20) membrane surface. A spreadability of the solution was defined as a characteristic of solutions with higher viscosity than water, which determines how the liquid wets the surface of the chosen substrate. This is crucial due to the hydrophobic surface of the PS20 base membrane onto which a matrix layer with hydrophilic properties is poured. A 2 wt.% PVA solution exhibited lower viscosity than the 4 wt.% and 5 wt.% solutions, resulting in the highest spreadability. In further studies, we decided to use a 5 wt.% aqueous PVA solution, which served as the base for creating the polymer matrix. Its viscosity was higher, and the slightly lower water content contributed to faster evaporation from the system. Due to the crosslinking of PVA, the active layer did not wash away during pervaporation tests. At the same time, no membrane damage or loss of its transport properties was observed.

Crosslinking agents covalently bind PVA chains to form very large macromolecules, in the form of insoluble gel in the active layer. In membrane preparation studies, five types of crosslinking agents were used. Crosslinking was carried out using appropriate amounts of 50 wt.% aqueous solutions of glutaraldehyde (GA), and/or citric acid (CA), or 40 wt.% aqueous solution of glyoxal, or aqueous solutions of tannic acid or tartaric acid, dissolved in situ.

The chemical compositions of the components used for PVA/PSf/PES membrane preparation are presented in Table 3 where the content of each component in relation to the entire volume of the mixture is provided. Initially, glutaraldehyde (GA) was used for crosslinking (Figure 3), but due to carcinogenic properties of GA, glyoxal (Figure 4) and citric acid (CA) (Figure 5) began to be used. Citric acid, due to its chemical structure (a four-functional carboxylic acid containing 3 reactive carboxyl groups and one reactive hydroxyl group), serves as an excellent crosslinking agent, creating spatially crosslinked structures with the polymer [89]. Interesting alternatives to citric acid are tartaric acid and tannic acid (see Figure 1) found in natural tannin. This compound was used due to its branched chemical structure and a large number of phenolic groups. Tannic acid in reaction with GA can create a more branched spatial chemical structure, with larger “structural channels” between the crosslinked PVA chains. These channels allow faster transport of permeating components to the other side of the membrane.

### 3.2. Characteristics of the Membranes

Membranes analyses were conducted both on the surface of the active layer matrix (top layer) and throughout the entire volume of the membrane (cross-section). Surface studies included measurements of contact angle (*δ*) and SEM imaging and analysis. Volume studies involved determination of a swelling degree (S), membrane density (ρ_d_) and thickness (*D*). The results are summarized in Table 4.

The thickness of the supporting layer of the produced PV membrane is the thickness of the commercial PS20 membrane, which has a repeatable average dimension of 101.2 ± 1 µm. The thickness of the active layer, responsible for the efficiency of the desalination process, ranged from 7.6 to 35.4 µm. The average thickness of this layer for the membranes analyzed in Table 3 was 16.3 ± 1 µm, which is consistent with the literature data.

The active layer of the pervaporation membrane is nonporous and its permeability results from the existence of space at the molecular level of the polymer chain structure. Therefore, in the case of PV membranes, the thickness of the separation layer cannot be discussed in combination with pore intrusion.

Due to the integrity of the active and support layers, swelling measurements were performed for the entire produced PV membranes. As shown in Table 4, the swelling degree (*S*) of the laboratory-scale produced and tested membranes ranged from approximately 49% to 68%. Due to the significant hydrophilicity of the active layer of these membranes, their swelling degree was higher compared to the PS20 substrate membrane (*S* = 41%). Swelling, in the case of a component flowing through the cross-section of several different materials, affects the phenomenon of transport resistance in the entire membrane. The greater the swelling of individual materials, the longer the path of the permeating component, which significantly affects the total membrane resistance. Due to the observed swelling phenomenon of the hydrophobic backing membrane, it was justified to examine the material in its entirety. After calculating, in accordance with the methodology included in point 2.2.1, the degree of swelling of the hydrophilic active layer, it turns out that due to its thickness, it had a 28% impact on the swelling of the entire PV membrane produced. As it can be seen from Figure 6 and from results presented in Table 4, the surface PVA layer was nonporous, so it is quite obvious that the overall swelling degree was higher than for the support layer.

This is supported by the measured contact angle values of the membrane surfaces, which were mostly below 45°, indicating considerable hydrophilicity of the membranes. The contact angle of the applied active layer composed of cross-linked PVA for all membranes was lower compared to the PS20 substrate membrane, as shown in Table 4. The commercial PS20 membrane, serving as the ultrafiltration substrate for the newly developed pervaporation membranes, is a typical hydrophobic membrane. Figure 7 illustrates the relationship between the swelling degree and the contact angle of the membranes.

From the graph in Figure 7, it can be observed that as the contact angle of the membranes increased, the degree of swelling slightly decreased due to reduced hydrophilicity of the membranes. A higher contact angle indicates lower hydrophilicity, resulting in reduced transport capabilities of the membrane due to decreased affinity of the membrane material to the solvent, which in this case is water.

The thickness of the pervaporation membranes varied. The thickest membrane was M69, which was produced with the addition of the detergent Rokanol L4P5, known for its high viscosity detergent solution, potentially leading to the formation of a thicker active layer of the membrane. The thinnest membranes were M56, M61, and M88, as indicated in Table 4. According to the results shown in Figure 6 and Table 4, the PVA active layer was nonporous, so the overall density was higher than the support layer PS20 (1.02 g/cm^3^). Our research showed that cross-linking of PVA with glutaraldehyde (GA) (M52–55, M69, M88, M121–122) gave a material with the slightly higher density (1.11 g/cm^3^) compared to membranes containing PVA layer cross-linked with citric acid (CA) for which the density was 1.09 g/cm^3^ (M56, M63, M67, M75, M86, M120). The average *J_p_* for PV membranes with an active layer cross-linked with CA was higher than for GA.

SEM images of selected membranes in Figure 6 reveal the surface and internal structure of the active layer. In some SEM images, polyester fibers are visible at the bottom of the membranes (Figure 6b,c). As mentioned earlier, the membranes produced for the PV process consist of three layers. The SEM images show the cross-section of the PS20 substrate membrane onto which the active polymer layer from cross-linked PVA was applied. The SEM images display a relatively uniform, smooth surface of the active layer of the membranes. This layer does not exhibit characteristics typical of porous membranes. This seems to be an evidence of achieving asymmetric membranes with the dense active layer suitable for pervaporation processes. Minor contaminants are visible on the surface, which may have occurred during membrane storage (Figure 6c,d), or as elements of fouling deposited on the membranes during the desalination process (Figure 6a,b).

The properties of the produced membranes were compared to those of the commercial PERVAP 4510 membrane, designed for alcohol dehydration processes. This membrane was utilized by the authors in the PV desalination process and its study was the subject of another article [83]. PERVAP 4510 is the polymer membrane based on PVA deposited on the polysulfone substrate, similar to the membranes developed by the authors. In Figure 6e, the smooth surface layer representing the active layer of the membrane is visible. Comparing PERVAP 4510 to the membranes developed in our study, it can be observed that their active layer surfaces are similar except for the visible cracks on the surface of the PERVAP membrane. This phenomenon occurred during the preparation of the SEM sample, although it is unclear why similar cracks were not observed on other membranes. The processing procedure was identical in all cases.

A photograph in Figure 8 reveals the interesting structure of the M61 membrane containing elements of PEG 200 structure. Under appropriate magnification, microstructures resembling cracks are visible. This may be a result of the dried hydrogel of the membrane containing PVA and PEG 200. It is also important to emphasize that these surface cracks did not negatively impact the efficiency or selectivity of this membrane. It demonstrates that the hydrogel, after swelling in water, enables a selective and highly efficient transport of the separated components. The incorporation of PEG 200 into the cross-linked polymer system has improved the process efficiency, as evidenced by the results obtained for the M61 membrane.

### 3.3. The Influence of Crosslinking Agents on Membranes Transport Properties

The produced PVA/PSf/PES membranes were subjected to water desalination tests using the PV method, determining their permeate flux and a retention rate, representing efficiency and selectivity. The results are shown in Figure 9. As indicated by the data in Table 3 and Figure 9, a twofold decrease in glutaraldehyde (GA) concentration resulted in over a fourfold increase in permeate flux, with a simultaneous increase in the selectivity of the water desalination process for membranes M52 and M54, respectively. Conversely, a twofold increase in citric acid (CA) concentration led to more than a twofold decrease in the permeate flux for membranes M56 and M63, respectively. The use of a mixture of crosslinking agents, GA and CA, for membrane M57, resulted in decreased efficiency compared to membranes crosslinked solely with CA, regardless of its amount—M56 and M63. Comparing the properties of membranes M56 and M63 with those of membrane M52, which was obtained with a larger amount of GA used for PVA crosslinking, shows lower performance for membrane M52 compared to the membranes crosslinked with CA (M56 and M63).

The addition of PEG 200 can participate in crosslinking reactions with GA and/or CA due to its chemical structure, resulting in the formation of new hydrophilic hydrogel structures in the polymeric active layer of modified membranes. The use of PEG 200 (with a low number-average molecular weight of approximately 200 g/mol) enabled the formation of a PVA hydrogel with structural elements of GA, CA, and PEG 200. The obtained hydrogel in the active layer of the PV membrane matrix resulted in better mechanical properties of the whole membrane and an increased degree of swelling, ensuring favorable water transport through the membrane [90]. The application of PEG 200 caused a significant increase in the permeate flux while maintaining the high degree of retention (for membrane M61). For membranes with PEG, it was observed that increasing the amount of CA in membrane M67 negatively affected the efficiency and selectivity of the desalination process compared to membrane M75 (for constant amounts of PEG 200). An excessive amount of the crosslinking agent CA in the hydrogel does not result in complete crosslinking of PVA but mainly grafts it onto the PVA chains. However, membranes modified with PEG 200 and GA (M121 and M122) did not exhibit the high permeate flux. Conversely, increasing the amount of PEG 200 (from 0.1 mL to 0.2 mL) while simultaneously reducing the amount of CA (from 0.4 mL to 0.2 mL) resulted in an approximately 60% increase in the permeate flux in the case of membranes M67 and M120. In Table 5 are presented the results of the obtained permeate fluxes and salt retention degrees.

The highest selectivity of the desalination process was achieved for membranes M61 and M102, while the highest desalination efficiency was achieved for membrane M102, crosslinked only using glyoxal, with a permeate flux of 17.5 kg/(m^2^h). This value is satisfactory compared to the literature data shown in Table 2.

### 3.4. The Influence of the Type of a Surfactant on Membrane Transport Properties

To produce the active layer of PV membrane (applied to the PS20), membrane-forming solutions with the addition of the popular surfactant Tween 20 or two polyethers (Rokopol 30P10 and Rokanol L4P5) or detergent LC were used. Tween 20 is used in numerous chemical applications, detergent LC commonly used in household chemistry. The surfactants, due to their high viscosity and ability to reduce surface tension, cause better spreadability of the membrane-forming solution, resulting in more uniform coating of the PS20 membrane substrate.

In Table 3 are summarized the chemical compositions of the individual membranes, including the type of the surfactant and the amounts of each component. Next, pervaporation desalination experiments were conducted on the newly prepared membranes. A summary of the influence of the type of the surfactant on the efficiency and selectivity of the produced membranes is presented in Figure 10.

The analysis of the surfactant influence (see Table 3 and Figure 10) indicates that membranes M63 and M86, crosslinked with CA (0.2 mL and 0.4 mL, respectively) in the presence of detergent LC (0.4 mL and 0.2 mL, respectively), exhibited very good efficiency and selectivity. In the case of membrane M69 containing Rokanol L4P5 after conducting a series of three PV tests over 2 h, it was observed that the membrane became fouled, which was not observed for other membranes. Contaminants were visible on the membrane surface, suggesting possible leaching of the surfactant. As an antifouling substance, the surfactant should effectively prevent the deposition of contaminants on the membrane surface. Membrane M55, prepared with the addition of Rokopol 30P10 detergent, also initially showed high flux values and high retention values. However, after 2 h of operation, its selectivity drastically decreased. This decrease in selectivity could be due to fouling or other operational factors that affected the membrane’s performance.

Detergent LC and Tween 20 contain emulsifying agent—Polysorbate 20 in their composition. Therefore, its use alongside Tween 20 seemed beneficial. This was confirmed by the efficiency and selectivity results obtained for membranes M86 and M88, as shown in Figure 11, and for membranes ranging from M63 to M122, presented in Figure 10. The performance of membranes containing detergent LC did not differ significantly from those containing Tween 20. In the Table 4 are presented the achieved permeate flux values and salt retention degrees for membranes with various surfactants.

### 3.5. The Influence of Cross-Linking Density on Membranes Transport Properties

The cross-linking density was calculated based on the degree of swelling and tensile strength, according to Equations (2) and (3) in this study. All membranes exhibited a higher cross-linking density compared to the PS20 substrate membrane. The cross-linking density of the membranes influenced the permeate flux obtained. From the Figure 11, it can be inferred that using glutaraldehyde (GA) for cross-linking PVA resulted in membranes with lower permeate flux and relatively minor changes in cross-linking density values. Adding components containing GA and polyethylene glycol (PEG 200) to the mixture increased the cross-linking density (seen in membrane M122 compared to membrane M52). Furthermore, comparing membranes M121 and M122, which contain the same amount of PEG 200, but different amounts of GA (M121 with reduced GA), there was observed a significant increase in the cross-linking density with a simultaneous decrease in the permeate flux.

A comparison of the properties of membranes M102 and M107 containing glyoxal showed an increase in cross-linking density with the higher amount of glyoxal in the membrane-forming solution (M102). An interesting phenomenon was observed with membranes M120 and M67, which were cross-linked using CA and PEG 200. Decreasing the amount of CA while increasing the amount of PEG 200 did not affect the cross-linking density of the membrane M120, suggesting that it is possible to reduce the cross-linking agent, while using a gelling agent (PEG 200). Similarly, no change in the cross-linking density was observed when reducing the amount of PEG 200, while keeping the amount of CA constant (in the case of membranes M120 and M75). This is intriguing because the permeate flux doubled for the membrane M75, as compared to the membrane M120.

For membranes containing two types of the cross-linking agents, CA and GA (M57, M61), adding PEG 200 to the component system increased the cross-linking density, and in the case of M61, it also caused the increase the permeate flux. It is possible that addition of PEG 200 was responsible for the increase of the hydrophilicity of the membrane, which positively influenced the obtained permeate flux (*J_p_*). It can be speculated that using different cross-linking agents and an additional compound forming a hydrogel with PVA may result in the formation of numerous channels (inside of membranes) that facilitate easier transport of separated components.

For all studied PV membranes were obtained high values of a salt rejection (*R* > 99%), but not only for membranes modified with addition of tartaric acid (M115) or tannic acid (M116) were observed relatively low values of permeate flux (*J_p_*)—see all results in Table 5. So far we did not find any correlation between *J_p_* and crosslinking density (*υ*) (and also contact angles, *δ*). Chemical compositions of water PVA solutions with different additives were different. So it was difficult to compare obtained results. More experimental results could help to explain such efficiency of membranes M115 and M116. These studies will be conducted in a future.

## 4. Conclusions

An experimental study of the composition of the active layer applied to the polyester-polysulfone support membrane to determine the most favorable, in terms of selectivity and efficiency, for pervaporation desalination of water is presented. The composition of the active layer was modified by varying the proportion of the main PVA polymer, crosslinking agents and surfactants.

On a laboratory scale, an active PVA layer was prepared on a surface of a commercial PS20 ultrafiltration membrane. The PVA/PSf/PES membrane was intended for a seawater desalination. An effect of the type and concentration of cross-linking substances and the influence of the type of surfactant on the characteristics of membranes were examined. Membrane separation tests were carried out using pervaporation process (PV).

The conducted studies indicated that the useful concentration of PVA, the polymer forming the active layer of the membrane, was 5% by weight. It was demonstrated that citric acid, which is safe for humans and the environment, serves as a better cross-linking agent compared to glutaraldehyde, which has carcinogenic properties. The concentration of the cross-linking agent also has practical significance. To obtain good membranes for the desalination via pervaporation (high efficiency and selectivity), an addition of certain and relatively small amounts of cross-linking agents (in the presence of sulfuric acid as a catalyst) to active PVA layer was sufficient. The use of a 40 wt.% aqueous solution of glyoxal for cross-linking yielded the best results in terms of efficiency and selectivity of the PV process. The application of tartaric acid and tannic acid resulted in membranes with an appropriate degree of the cross-linking, but they did not achieve outstanding efficiency compared to other membranes. Surfactants used as anti-fouling agents improved the hydrophilic properties of the membranes, simultaneously reducing the fouling, which is the cause of permeate flux reduction. The use of a membrane-forming solution containing PEG 200, after cross-linking PVA with GA or CA, resulted in the formation of PVA hydrogels with elements of PEG 200 and GA (or CA) structures. It was demonstrated that employing a mixture of GA and CA for cross-linking the hydrogel membrane M61 had the most favorable impact on the efficiency and selectivity of the PV process. The use of a the hydrogel membrane obtained by cross-linking PVA and PEG 200 with CA (M75) also led to the high permeate flux, compared to membranes modified by cross-linking PVA and PEG 200 with GA.

Dry phase inversion method used for obtaining pervaporation membranes for water desalination was found to be advantageous due to its simplicity of preparation. The characterization of the membranes showed the presence of the active layer deposited on the polyester-polysulfone PS20 substrate membrane, and the effectiveness of its modification in terms of the hydrophilicity, swelling degree, thickness, and membrane density. SEM images confirmed the deposition of the dense active layer, using the dry phase inversion method on the microporous PS20 substrate. The properties of the prepared membranes were compared to the commercial PERVAP 4510 membrane, which is also the PVA-based membrane deposited on the polysulfone substrate. The SEM images confirmed similarities in the active layer, and the achieved retention and efficiency results suggest that the developed by us membranes could successfully be used for water the desalination of saline water via the PV method. The permeate flux for the PERVAP membrane in previous studies was maximally around 2 kg/(m^2^ h), whereas for PVA/PSf/PES—the M102 membrane developed in this study, the flux was ~18 kg/(m^2^ h), confirming that the obtained membranes can compete with commercial PV membranes. Further studies on membrane preparation should allow for the preparation of significantly more efficient and highly selective membranes for the PV process.

## Figures and Tables

**Figure 1 membranes-14-00213-f001:**
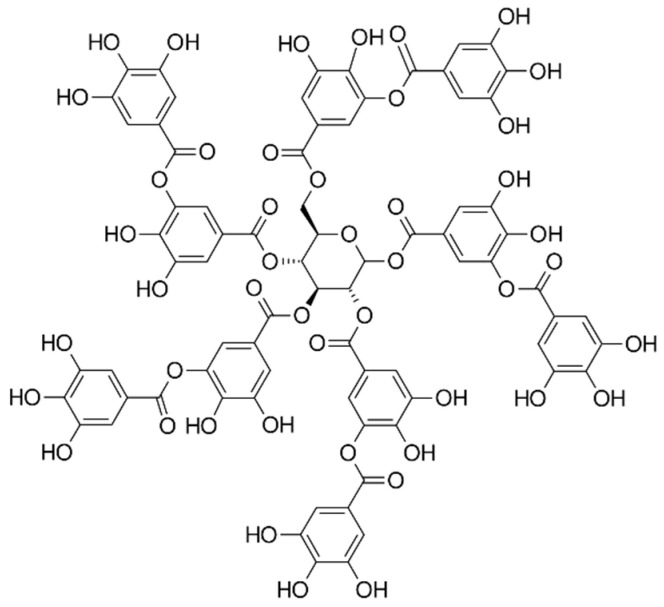
A structural formula of tannic acid.

**Figure 2 membranes-14-00213-f002:**
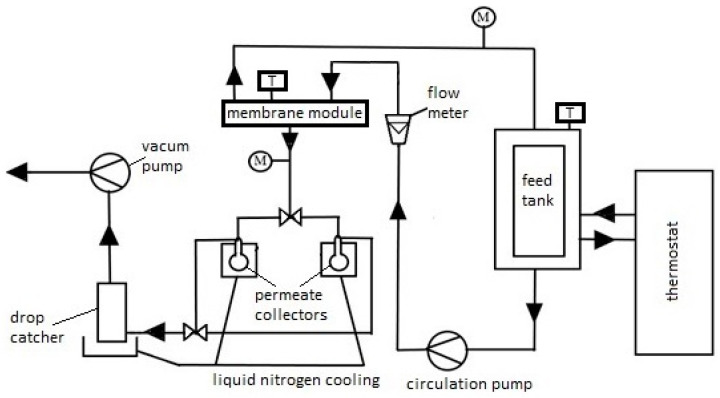
Diagram of laboratory apparatus for water desalination using the PV process (M—pressure gauge, T—thermometer), arrows show the direction of feed and permeate flow.

**Figure 3 membranes-14-00213-f003:**
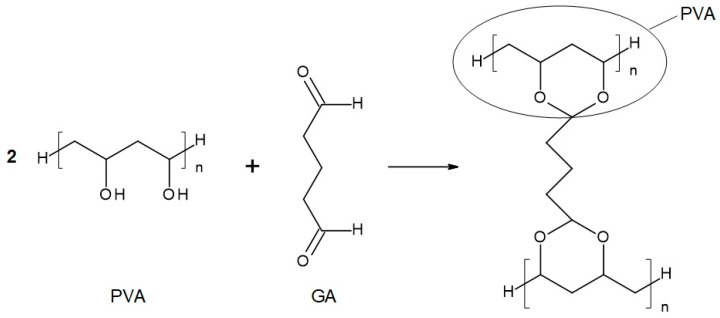
Reaction scheme of the crosslinking reaction of polyvinyl alcohol (PVA) with glutaraldehyde (GA).

**Figure 4 membranes-14-00213-f004:**
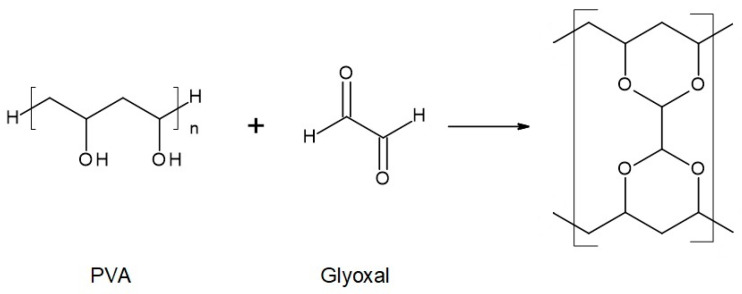
Reaction scheme of the crosslinking reaction of polyvinyl alcohol (PVA) with glyoxal solution.

**Figure 5 membranes-14-00213-f005:**
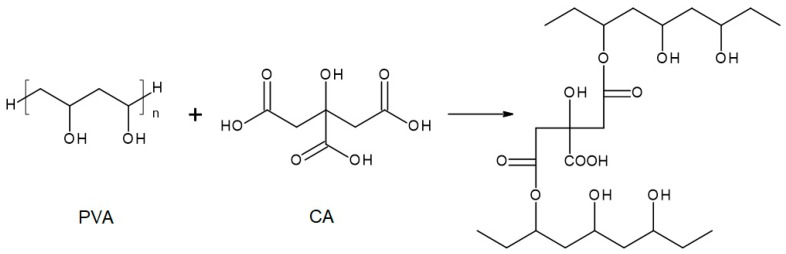
Reaction scheme of the crosslinking reaction of polyvinyl alcohol (PVA) with citric acid (CA)—under assumption that only 2 COOH groups of CA reacted with PVA.

**Figure 6 membranes-14-00213-f006:**
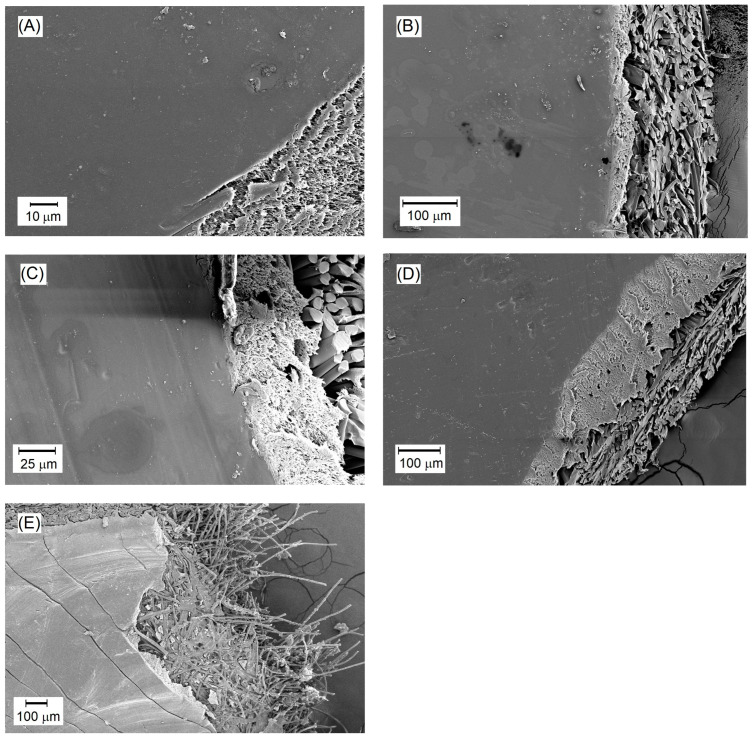
SEM images of the surfaces of selected PVA/PSf/PES membranes: (**A**) M56; (**B**) M67; (**C**) M75; (**D**) M86; (**E**) PERVAP 4510.

**Figure 7 membranes-14-00213-f007:**
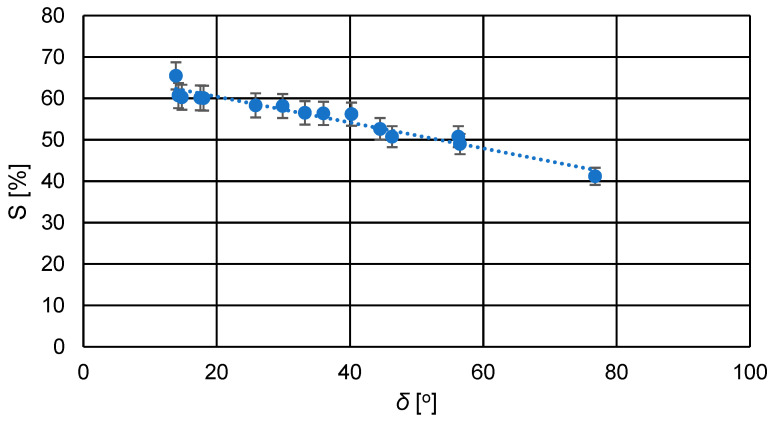
Relationship between the swelling degree (*S*) and the contact angle (δ) for the prepared PVA/PSf/PES membranes.

**Figure 8 membranes-14-00213-f008:**
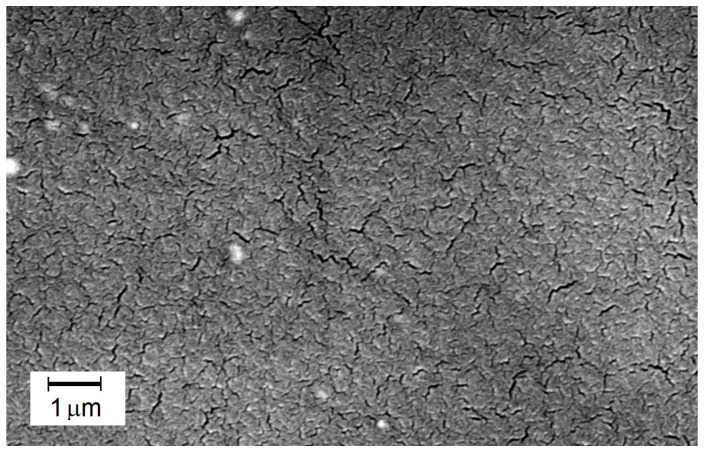
SEM image of the surface of the active layer of the M61 membrane containing PEG 200.

**Figure 9 membranes-14-00213-f009:**
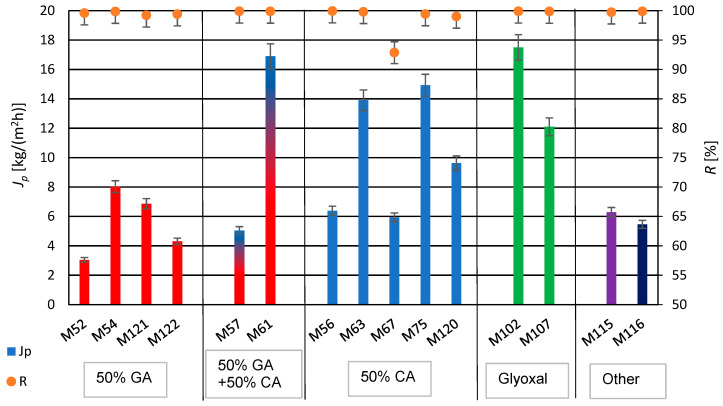
Summary of permeate flux (*J_p_*) and retention degree (*R*) values after the water desalination process by PV method (*T* = 60 °C, *Q_f_* = 60 dm^3^/h) for membranes crosslinked with selected crosslinking agents.

**Figure 10 membranes-14-00213-f010:**
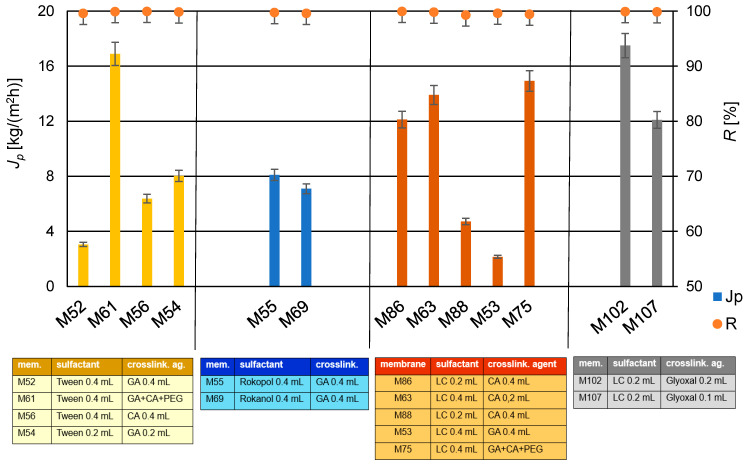
The summary of the influence of the type of the surfactant on the efficiency (*J_p_*) and selectivity (*R*) of selected PVA/PSf/PES membranes for the water desalination process by PV (*T* = 60 °C, *Q_f_* = 60 dm^3^/h).

**Figure 11 membranes-14-00213-f011:**
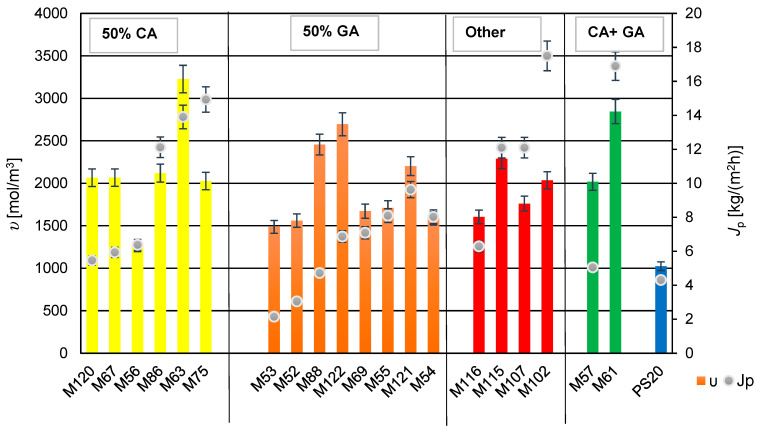
The relationship between cross-linking density (*υ*) and permeate flux of developed membranes (*J_p_*). The bars on the graph indicate the cross-linking density—left axis; Dots indicate the permeate stream—right axis.

**Table 1 membranes-14-00213-t001:** A review of the literature on the formation of pervaporation membranes and the most important achievements during their creation.

Possible Application	Membrane Manufacturing Method	Materials Forming the Membrane	Results	References
PV		PVA, fillers: GO, zeolites, silica	excellent transport properties; resistance to scaling or/and fouling	[25]
DCPV	phase inversion	PVA/PVDF/PTFE	three-layer membrane; high efficiency (*J_p_*, *R*); anti-fouling properties	[46]
PV	in situ polymerization	PVA, silica, PTES, DEDPS as silica precursors		[47]
PV	sol-gel	silanes: TEOS, MTES (acted as a precursor for micropores in the transport layer)	salt retention rate of 99%	[48]
PV	sol-gel	PVA, MA, TEOS (as the silica precursor)	highly homogeneous hybrid membranes; MA led to formation of amorphous regions, which increased the diffusion of water through the membrane	[49]
PV		PVA/MA, silica	the use of silica reduces the swelling phenomenon of membranes	[50]
PV	sol-gel and carbonization at 175 °C	TEOS, CA and NH_4_OH (to enhance their hydrostability)	xerogels could be useful in the preparation of PV membranes	[51]

GO—graphene oxide; DCPV—direct-contact pervaporation; PTES—phenyl(triethoxy)silane; DEDPS—(diethoxy)diphenylsilane; TEOS—(tetraethoxy)silane; MTES—methyl(triethoxy)silane; MA—maleic acid.

**Table 2 membranes-14-00213-t002:** A review of the process parameters of different composite membranes prepared for water desalination in PV process.

Feed	Membrane Manufacturing Method	Materials Forming the Membrane	Results		References
highly saline water		PVA, 2% wt laponite (group of aluminosilicates)	10% wt. NaCl; *T* = 40–70 °C	*R* = 99,9%	[60]
highly saline water	chemical cross-linking	PVA dispersions, GA, 7% wt. laponite	*T* = 40 °C	*R* = 99.98%	[61]
saline water		zeolites with CHA and FAU structures	*T* = 90 °C	*J_p_* = 40–51 kg/(m²h)	[52]
saline water		PVA, MCWT	multi-walled membranes with 0.3% wt. MCWT	*R* = 98.8%	[62]
saline water		hydrated cellulose and cellulose diacetate		*J_p_* = 6–7 kg/(m^2^h)	[63]
saline water	interfacial polymerization	filler: quantum dots	TFN membranes; *T* = 70 °C	*R* = 99.98% *J_p_* = 23.2 kg/(m²h)]	[64]
saline water	wet phase inversion	GO	membranes doped with GO	*R* = 99.9%; *J_p_* = 36.1 kg/(m²h)	[65]
saline water	Intercalation NH_2_-POSS in GO layer	oligomeric NH_2_-POSS), GO, GA	stable performance over a 24h desalination	*R* = 99.8% *J_p_ =* 112.7 kg/(m²h)	[66]
concentrated inland brine		PP, PVA with GO and chitosan	structure defects,	*R* = 99.99% *Jp* = 12.8–30.5 kg/(m^2^h)	[67]
		PP, PVA with GO and chitosan, DBS	lack of membrane stability	low *R*	
saline water	electrospinning	PVA, aliphatic compounds containing sulfonic groups	membranes 0.73 μm thick for PV and MD; hydrostable and mechanically strong	*R* = 99.7% *J_p_* = 234.9 kg/(m^2^h)	[31]
saline water		PAN, PVA with cellulose nanofibers	TFC for 20% wt. NaCl; *J_p_* = 103.1 kg/(m^2^h)	*R* = 99.8% *J_p_* = 238.7 kg/(m^2^h)	[68]
saline water	solution casting	thin-layer lignin	TFC	*R* = 99.95% *J_p_* = 18.5 kg/(m^2^h)	[69]
wastewater desalination		PAN UF membrane, PVA crosslinked with PMDA	PVA layer of 2 µm thickness; crosslinked with 20% wt. PMDA, 100 °C for 2 h	*R* = 99.98% *Jp* = 32.26 kg/(m²h) at *T* = 70 °C, 3.5% wt. NaCl	[70]
desalinating seawater, saline water, treating concentrate from RO		SA-PVA/PAN membrane PVA, SA, commercial PAN UF membrane	performance 120 h, *p* = 100 Pa, *T* = 70 °C, 3.5% wt. NaCl	*R* = 99.8% *Jp* = 27.9 kg/(m²h)	[71]
			10% wt. NaCl	*R* = 99.8% *Jp* = 11.2 kg/(m²h)	
seawater desalination; highly saline water	reaction of CD-based neutral ions	HPAN, terephthaldehyde	treatment at low temperatures, hydrophilicities; gas permeabilities *T* = 25 °C	*R* = 99.8% *Jp* = 15.0 kg/(m²h)	[72]
saline water	spraying 0.6 μm layer of PVA onto a PSf, PSf prepared using NIPS	PVA/PSf UF membrane	3.5% wt. NaCl; *T* = 70 °C	*R* = 99.9% *Jp* = 124.8 kg/(m²·h),	[73]
			20% wt. NaCl; *T* = 70 °C	*Jp* = 71.3 kg/(m²·h)	

MCWT—carbon nanotubes; TFN—thin film composite membrane; GO—graphene oxide; NH_2_-POSS—aminopropylisobutyl) silsesquioxane; PP—polypropylene; DBS—2,5-diaminobenzenesulfonic acid; MD—membrane distillation; PAN—polyacrylonitrile; HPAN—hydrolyzed polyacrylonitrile; SA—sulfosuccinic acid; UF—ultrafiltration; PMDA—pyromellitic dianhydride; NIPS—non-solvent induced phase separation method; CD—cyclodextrins; PSf—polysulfone.

**Table 3 membranes-14-00213-t003:** Comparison of the individual chemical compositions of active layer-forming solutions in PV membranes, considering the impact of the crosslinking agent and surfactant.

Membranes Numbers	PVA 5 wt.% Solution	Crosslinking Agents	Surfactants	Other Ingredients
**M52**	4.0 mL	0.4 mL GA *	0.4 mL Tween 20	-
**M53**			0.4 mL LC	
**M54**		0.2 mL GA *	0.2 mL Tween 20	
**M55**		0.4mL GA *	0.4 mL Rokopol 30P10	
**M56**		0.4 mL CA *	0.4 mL Tween 20	
**M57**		0.2 mL CA * 0.2 mL GA *		
**M61**		0.2 mL CA * 0.2 mL GA *		0.1 mL PEG 200
**M63**		0.2 mL CA *	0.4 mL LC	-
**M67**		0.4 mL CA *		0.1 mL PEG 200
**M69**		0.4 mL GA *	0.4 mL Rokanol L4P5	-
**M75**		0.2 mL CA *	0.4 mL LC	0.1 mL PEG 200
**M86**		0.4mL CA *	0.2 mL LC	-
**M88**		0.4mL GA *		
**M102**		0.2 mL Glyoxal (40 wt.% aqueous solution)		
**M107**		0.1 mL Glyoxal (40 wt.% aq. solution)		
**M115**		0.4 g Tartaric acid		
**M116**		0.05g Tannic acid 0.4 mL GA		
**M120**		0.2 mL CA		0.2 mL PEG 200
**M121**		0.2 mL GA		0.1 mL PEG 200
**M122**		0.4 mL GA		0.1 mL PEG 200

* 50 wt.% aqueous solutions of GA and CA were used.

**Table 4 membranes-14-00213-t004:** Comparison of characteristic parameters for the prepared PVA/PSf/PES membranes. Symbols mean: contact angle (*δ*), swelling degree (*S*), thickness (*D*) and density (*ρ_m_*) of the whole PV membrane.

Membranes Numbers	*δ* [°]	*S* [%]	*D* [µm]	*ρ_d_* [g/cm^3^]
**M52**	36.0	56.38	108.8	1.11
**M53**	44.5	56.20	126.0	1.12
**M54**	46.3	50.76	111.2	1.10
**M55**	14.8	60.29	118.2	1.10
**M56**	56.3	48.99	109.8	1.11
**M57**	40.2	52.62	125.2	1.13
**M61**	34.1	57.48	109.8	1.12
**M63**	13.9	60.69	120.4	1.07
**M67**	47.4	49.01	115.0	1.08
**M69**	29.9	58.17	136.6	1.09
**M75**	19.5	60.06	125.0	1.09
**M86**	34.0	56.50	122.8	1.09
**M88**	14.3	60.28	111.2	1.10
**M102**	17.6	60.14	118.0	1.10
**M115**	33.2	58.31	116.0	1.10
**M107**	21.2	58.12	111.4	1.06
**M116**	25.9	65.45	119.8	1.10
**M120**	30.0	58.66	110.6	1.12
**M121**	27.7	59.13	116.8	1.11
**M122**	23.0	59.98	111.8	1.11
**PS20**	76.8	41.18	101.2	1.02

**Table 5 membranes-14-00213-t005:** The summary of the results of the PV process (permeate flux and salt retention degree) for membranes containing various surfactants and cross-linked with different crosslinking agents.

Membranes Numbers	Crosslinking Agents	Surfactants	$J_{p}\;$ $\left[\frac{kg}{m^{2}h}\right]$	*R* [%]
**M52**	GA *	Tween 20	3.05	99.59
**M53**		LC	2.15	99.61
**M54**		Tween 20	8.03	99.85
**M55**	GA *	Rokopol 30P10	8.10	99.73
**M56**	CA *	Tween 20	6.38	99.94
**M57**	CA * + GA *		5.05	99.90
**M61**			16.90	99.89
**M63**	CA *	LC	13.91	99.81
**M67**			5.94	92.87
**M69**	GA *	Rokanol L4P5	7.09	99.58
**M75**	CA *	LC	14.93	99.43
**M86**	CA *		12.12	99.93
**M88**	GA *		4.72	99.28
**M102**	Glyoxal **		17.50	99.90
**M107**			12.10	99.87
**M115**	Tartaric acid		6.29	99.74
**M116**	Tannic acid + GA *		5.46	99.89
**M120**	CA *		9.63	99.01
**M121**	GA *		6.87	99.21
**M122**			4.31	99.41

* 50 wt.% aqueous solutions of GA and CA were used; ** 40 wt.% aqueous solution of glyoxal.

## Data Availability

The raw data supporting the conclusions of this article will be made available by the authors on request.

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
