# Peer review of "The Efficiency of Polyester-Polysulfone Membranes, Coated with Crosslinked PVA Layers, in the Water Desalination by Pervaporation"

_membranes, 2024, doi:10.3390/membranes14100213_

Round 1
Reviewer 1 Report
Comments and Suggestions for Authors
In this research,the effects of various crosslinking agents and surfactants on the desalting properties of PVA composite membranes were investigated. Multiple membranes were prepared and subjected to characterization and separation performance testing in this study. However, the discussion is generally weak and lacks depth, with some performance aspects not adequately addressed. It is necessary to strengthen the analysis. A relatively comprehensive review was made at the beginning of the paper, but before reference 57, the separation properties of membranes were not involved, which made it difficult for us to make a comparison in performance. The author should make some additions. In addition, the manuscript has the following issues to discuss:
1、The title fails to accurately convey the essence of the paper, thus it is recommended to make modifications.
2、In the experiment, Qf=60 dm3/h, the flow rate seems to be a little small, is it possible to cause a serious concentration polarization and affect the accuracy of the experimental results?
3、SEM image of the surface (Fig 10), the magnification ratio should be consistent for all the figures; For the convenience of comparison and inspection of the poro intrusion, give the SEM image of the membrane cross section is better.
4、 The film thickness discussed in this paper specifically refers to the thickness of the composite film, without any further elaboration. In fact, it is more meaningful to discuss the thickness of the separation layer in combination with the pore intrusion. Because the thickness of the support layer is not uniform, there may be a large error in comparing the thickness of the separated layer by measuring the thickness of the entire composite film.
5、When measuring the swelling degree, pure PVA film should be used instead of composite film.
6、Similarly, when measuring membrane density, pure PVA membranes should be used instead of composite membranes. Due to the large porosity of the support layer, a small change in porosity will cause a large difference in density, which will mask the change in PVA density. It makes sense to discuss the effects of crosslinkers and surfactants based on accurate density measurements. Also, there is very little discussion about density, and this sentence doesn't make much sense:“The densities of the produced membranes were also higher compared to the PS20 substrate membrane. This indicates that the active layer caused changes in the properties of the substrate membrane.”
7、About the calculation of crosslinking degree, equation 4 and 5 can not be found in this manuscript.
8、In line 746,"The application of tartaric acid and tannic acid resulted in membranes with an appropriate degree of the cross-linking, but they did not achieve outstanding efficiency compared to other membranes." What is the range of appropriate crosslinking degree, are there any guidelines?
9、line 55:"The mechanism of mass transport in the pervaporation process is based on the dissolution–diffusion model. ***with a solid active layer in which the adsorption and dissolution of the membrane components take place, followed by the diffusion of the evaporated components through the polymer material of the membrane." In fact, the model should be solution-diffusion model, and the process shoude be solution-diffusion-dissolution.
10、line 85: "Phase inversion involves pouring the membrane-forming solution onto a previously prepared composite polymer primer. Subsequently, chemical cross-linking takes place***". That's not an accurate statement. In fact, the phase inversioin dose not nesesarily occur crosslinking.
Comments on the Quality of English LanguageGood
Author Response
Thank you for your valuable comments. We have made revision and responded accordingly, please see the attachment.

Reviewer 2 Report
Comments and Suggestions for Authors
In this manuscript, the authors modified PSf membrane with a PVA hydrogel layer and then characterized it. The results and discussion are logical and reasonable. I think it can be accepted after a major reversion.
1. The standard diversion in Figures 6, 7, and 9 is missing.
2. To better compare the membrane changes before and after modification, I strongly suggest the author use high-resolution SEM images for tested membranes (top-view and cross-section tests).
3. Line 488, the authors mentioned the membrane antifouling property. Where are the corresponding tested results? Such as surface static adsorption or dead-end (crossflow) filtration experiments.
4. The introduction section seems too much. I do not understand why the authors did such a long literature review about the fabrication of polymeric membranes in a "research article".
5. More characterizations need to be done for the synthesized membranes. For example, XPS analysis (also FTIR) to prove the crosslinking reaction happened. High-resolution SEM images to prove the successful coating and the detailed thickness of the PVA hydrogel layer. Membrane stability tests in the short and long term are also required.
Comments on the Quality of English Languageno
Author Response

(The authors gave the same response as above.)

Round 2
Reviewer 2 Report
Comments and Suggestions for Authors
Well done!
Comments on the Quality of English Languageno